# High-Pressure Processing of Traditional Hardaliye Drink: Effect on Quality and Shelf-Life Extension

**DOI:** 10.3390/foods12152876

**Published:** 2023-07-28

**Authors:** Bahar Atmaca, Merve Demiray, Gulsun Akdemir Evrendilek, Nurullah Bulut, Sibel Uzuner

**Affiliations:** 1Center Research Laboratory Application and Research Center, Mardin Artuklu University, 47420 Mardin, Türkiye; baharatmaca@artuklu.edu.tr; 2Department of Food Engineering, Faculty of Engineering, Bolu Abant Izzet Baysal University, Golkoy Campus, 14030 Bolu, Türkiye; demiraymerve@gmail.com (M.D.); nurullahbulut@gmail.com (N.B.); 3Department of Food Engineering, Faculty of Engineering, Izmir Institute of Technology, 35420 Izmir, Türkiye; sibeluzuner@iyte.edu.tr

**Keywords:** hardaliye drink, high hydrostatic pressure, optimization, Box–Behnken design, shelf-life extension

## Abstract

Hardaliye, as one of the oldest and lesser known traditional beverages, is produced using red grape pomace from wine production. This drink production is achieved through lactic acid fermentation, with the addition of sour cherry leaves and mustard seeds—either heat-treated, grinded, or whole—in various concentrations. Hardaliye has a very short shelf life; thus, efforts have recently been made to process hardaliye with novel processing technologies in order to achieve shelf-life extension. Therefore, the high-hydrostatic-pressure (HHP) processing of hardaliye was performed to determine its impact on important properties, including in microbial inactivation and shelf-life extension, with respect to a Box–Behnken experimental design. Maximum log reductions of 5.38 ± 0.6, 5.10 ± 0.0, 5.05 ± 0.2, and 4.21 ± 0.0 with HHP were obtained for *Brettanomyces bruxellensis*, total mesophilic aerobic bacteria, *Lactobacillus brevis*, and total mold and yeast, respectively. The processing parameters of 490 MPa and 29 °C for 15 min were found as the optimal conditions, with the response variables of an optical density at 520 nm and the inactivation of *L. brevis*. The samples processed at the optimal conditions were stored at both 4 and 22 °C for 228 d. While the non-treated control samples at 4 and 22 °C were spoiled at 15 and 3 d, the HHP-treated samples were spoiled after 228 and 108 d at 4 and 22 °C, respectively.

## 1. Introduction

Hardaliye is a traditional non-alcoholic beverage produced using red grape pomace from wine production. Fresh sour cherry leaves and mustard seeds at different concentrations—either fresh or heat-roasted, and in a whole or grinded form—are added. Depending on the sugar content of pomace, sugar addition is also possible. This traditional drink is produced through lactic acid fermentation performed at room temperature for 7–10 d. If fermentation is practiced at lower temperatures (10–15 °C), it can be extended up to 20 d. Hardaliye can be consumed fresh right after production, or it can be aged over several months; if it is aged, it may develop alcohol. Its characteristic red/burgundy color comes from grape pomace, whereas its very pleasant aroma is formed by grape pomace, sour cherry leaves, and mustard seeds [1,2]. It is traditionally produced and widely consumed, especially in the Thrace region of Turkiye.

Hardaliye is regarded as a non-dairy probiotic beverage because of the fermentation step during the production and its rich lactic acid microflora. Hardaliye has a high antioxidant capacity and phenolic substance content, with more than 15 individual phenolic substances, including trans-resveratrol, gallic acid, 2,5-dihydroxybenzoic acid, protocatechuic acid, and ABA, with the highest concentrations [3]. In addition to its functional and health-promoting effects, its consumption is recommended because of its beneficiary effect on the digestive system, prevention of coronary heart disease, and decreased serum homocysteine concentrations [2]. The mustard oil present in hardaliye also helps to heal disorders in the circulatory system, bronchitis, and a cold, and presents antimicrobial properties against some common pathogenic microorganisms. It is a good source of energy as 225 mL of hardaliye provides 29 carbohydrates and 170 kcal of energy. It can be consumed by people of all ages, including children, vegetarians, dairy-intolerant, hypertensive, and high-cholesterol individuals [4].

The grape fermentation process also helps to develop the nutritional value of hardaliye, whereas the functional and health-promoting properties are attributed to its ingredients of particular grapes, with a high phenolic content, and mustard seeds, containing etheric oils, allyl isothiocyanate, and sinigrin—a cinogenesis-suppressing agent. Moreover, hardaliye also helps to regulate the digestive system and was proved to be helpful in preventing coronary heart disease [2]. Even though hardaliye is a very special drink, its production volume is limited due to its short shelf life. The current practices of increasing its shelf-life are realized through the addition of sodium benzoate; however, this is not accepted by consumers, as concerns are raised over the adverse effects of sodium benzoate on human health. Thus, alternatives to sodium benzoate addition are in demand. Studies employing heat treatments provided shelf-life extension; however, the physical, health-promoting, and sensory properties of hardaliye were adversely affected. Except for ultrasonication (US) in the processing of hardaliye, with respect to the determination of changes in its physical, bioactive, and sensory properties with shelf-life extension, no studies have been performed.

A high hydrostatic pressure (HHP) is one of the most promising non-thermal processing technologies, with the application of high isostatic pressures ranging from 100 to 1000 MPa. HHP, being a less invasive non-thermal process, is classified as minimally processed, clean-label, and microbial-safe, accompanied by a superior quality. Foods are processed using HPP due to microbial and enzyme inactivation and the preservation of physicochemical, bioactive, and sensory properties with shelf-life extension, in addition to an increased extraction efficacy, modification of freeze–drying and textural properties, and size reduction. Both solid and liquid foods, including traditional drinks of a perishable nature with flexible packaging, which tolerate a certain level of compressive force, and without packaging at room temperature, in combination with heat treatment, or lower temperatures, even below the freezing point, are successfully processed via HPP [5,6,7,8,9]. HHP is the most developed and widely used non-thermal processing technology in the food industry.

The application of a high pressure in the range of 20–100 MPa with homogenization, known as high-pressure homogenization (HPH), is common for processing food products, especially juices and dairy beverages, with the main objectives of particle size reduction and increasing the emulsion stability by preventing coalescence phenomena and creaming. Moreover, the magnitude of the pressure has a direct effect on the cell disruption and recovery of intracellular bioactive compounds, which enables this technology to be used for food processing, with an improvement in food safety and shelf life [10,11].

Even though different foods have been successfully processed with HHP, limited studies are available on the effectiveness of HHP in traditional drinks, and no studies have been conducted on the HHP processing of the hardaliye drink. Thus, the objectives of the study were to determine the effects of the processing parameters on the physical, bioactive, and sensory properties of hardaliye, with the inactivation of endogenous and spoilage microflora; optimize and validate HHP-processing parameters; and quantify the shelf-life extension of hardaliye at both 4 and 22 °C.

## 2. Materials and Methods

### 2.1. Materials

#### 2.1.1. Materials and Equipment

All solutions were prepared in calibrated glassware. Finnpipette-brand micropipettes (±5 µL) (Sigma Aldirch, Darmstadt, Germany) were used in all analyses. An Orion perpHect logR meter from InoLab WTW, Weilheim, Germany was used for pH measurement; a 507-1 model refractometer from Nippon Optical Works Co., Ltd., Nagano-Ken, Japan was used to measure the total soluble solid; a Sension 5 model conductivity meter (HACH, CO, ABD) was utilized to measure conductivity; a Hunter Color Flex spectrophotometer (Hunter Associates Laboratory Inc., Reston, VA, USA) was used to measure color *L**, *a**, and *b**; and a T80+UV/VIS model spectrophotometer (PG Instruments, Leicestershire, UK) was used to measure total phenolic substance content (TPSC), total antioxidant capacity (TAC), total monomeric anthocyanin content (TMAC), color density (IC), color tone (CT), yellow color tone (YCT, OD_420_), blue color tone (BCT, OD_520_), red color tone (RCT, OD_620_), and color intensity. Samples were centrifugated via a Nüve NF800 Model centrifuge (Ankara, Turkiye). All samples were weighed via a Kern Brand Scale (Lohmar, Germany). An Mmert brand sterilizator (Art Teknik, Ankara, Turkiye) was utilized for the sterilization of glass Petri plates for microbiologic analyses, and an Alp Brand autoclave (Art Teknik, Ankara, Turkiye) was used to sterilize agars, broths, and other equipment, such as plastics and glassware. A Nüve ST30 model water bath (Ankara, Turkiye) was used for the samples to hold at 50 ± 2 °C. The vortexing of the samples was performed via a Heidolph MR3001 model vortex (Scwabach, Germany). All microbiological analyses were conducted in a sterile cabin (Art Teknik, Ankara, Turkiye)

HHP processing was performed with an HHP pilot plant processing unit by Avure, (Middletown, OH, USA), and vacuum packaging of the samples was conducted via a vacuum packaging machine from APACK Packaging Technologies (Istanbul, Turkey).

#### 2.1.2. Reagents

3,5-dinitrosalicylic acid (DNS), glucose, 2,2-diphenyl-1-picryl-hydrazyl-hydrate (DPPH), Tris-HCl, ethanol, Folin–Ciocalteu and sodium carbonate, gallic acid, KCl, sodium acetate, and tartaric acid were provided by Sigma Chemical Co. (Stockholm, Sweden). Potato dextrose agar (PDA), plate count agar (PCA), MRS agar, a YDP medium, and peptone were obtained from Fluka (Munich, Germany). The isolation and identification of endogenous microflora were conducted via API 20C and API50 CHB/E tests (bioMérieux, Inc., Durham, NC, USA).

#### 2.1.3. Samples

Fresh hardaliye samples produced by Cabernet-type grape pomace from wine production were kindly provided by Karlıbağ Hardaliye (Kırklareli, Turkiye). The samples in an amber-colored bottle were kept at refrigeration temperature until use.

### 2.2. Microbial Cultures

The hardaliye samples were left to spoil at room temperature for several days. After this, both API50 CHB/E and API 20C test kits were used to screen grown microorganisms in different media. Both *Lactobacillus brevis* and *Brettanomyces bruxellensis* cultures were identified based on their biochemical reactions on test kits. Isolated bacteria were then subcultured on an MRS agar slant, and incubated at 30 ± 2 °C for 48 h. The bacteria cultures were transferred to MRS agar from the saline solution (SS), and the plates were incubated at 30 ± 2 °C for 72 h. The grown culture from the MRS plates was suspended in SS, and then the cultures were collected via centrifugation at 3500× *g* for 10 min [12]. The cells were inoculated into the hardaliye samples at the final concentration of 10^5–6^ cfu/mL.

The yeast culture, after isolation, was transferred to the YPD medium and inoculated at 25 ± 2 °C for 7 d. After subsequent centrifugation at 3500× *g* for 10 min, the collected cells were inoculated into the hardaliye samples at the final concentration of 10^5–6^ cfu/mL [13].

### 2.3. High Hydrostatic Pressure

Vertical HHP equipment with a 2 L pressure vessel capacity using water as a pressure medium was utilized to process the samples. Samples with a 400 mL volume were vacuum packaged in plastic multilayer pouches (polyethylene–aluminum–polypropylene) before HHP processing. The average temperature increase and average pressure increase and decrease rate per a 100 MPa increase were recorded as 0.5 ± 0.2 °C, 0.5 min, and 0.2 min, respectively. Based on preliminary experiments, 200–500 MPa, 3–15 min (time after achieving the set pressure), and 4–22 °C with 15 variants of the HHP treatment were applied using a Box–Behnken design with the experimental order (Table 1).

### 2.4. Measurement of Physicochemical Properties

TSS (°Brix), pH, turbidity (NTU), and conductivity (mS/cm) of the hardaliye samples were measured. The titratable acidity of the hardaliye samples as equivalent to lactic acid was determined via the titrimetric method. A DNS reagent was used to determine the reducing sugar content with glucose used as a substrate.

The values of *L**, *a**, and *b** were recorded, and chroma (*C**), hue (*h*°), and total color difference (∆E) values were calculated from the color values. D65/10° as a simulated daylight color, a standard illuminant with a 10-degree viewing angle, was used for the color measurement. The light source used for the color measurement emitted radiant energy in the form of visible light, a minor portion of the electromagnetic spectrum, including ultraviolet, X-rays, radio waves, and infrared light in the range of 400–700 nm. The absorbance mode was utilized for the color measurements.

Moreover, IC; CT; and red (RCT), blue (BCT), and yellow (YCT) color indices as percent color components were calculated from the optical density (OD) measurements at 620, 520, and 420 nm, and reported as RCT (OD_620_), BCT (OD_520_), and YCT (OD_420_), respectively [14]. The absorbance values at 540 nm were recorded as the color intensity [15]. Change in the color parameters was thus estimated with the following:(1)$$C^{*}=\sqrt{a^{2}+b^{2}}$$
(2)$$h^{0}=\arctan\;(b/a)$$
(3)$$\Delta E=\sqrt{{\left(L_{0}-L\right)}^{2}+{\left(a_{0}-a\right)}^{2}+{\left(b_{0}-b\right)}^{2}}$$
(4)$$\mathrm{Color}\;\mathrm{tone}={OD}_{420}/{OD}_{520}$$
(5)$$\%{\mathrm{OD}}_{420}=\frac{{OD}_{420}}{IC}\times 100$$
(6)$$\%{\mathrm{OD}}_{520}=\frac{{OD}_{520}}{IC}\times 100$$
(7)$$\%{\mathrm{OD}}_{620}=\frac{{OD}_{620}}{IC}\times 100$$

Color intensity (IC) = OD_420_ + OD_520_ + OD_620_(8)


### 2.5. Measurement of Bioactive Properties

TAC (%) of the samples was quantified via the DPPH free-radical method. Hardaliye and a Tris-HCl tampon with a pH of 7.4 were mixed to obtain a homogenous mixture via vortexing at 2200 rpm for 6 min, and then 1 mL of the DPPH solution prepared in ethanol was transferred. The samples were retained for 20 min at room temperature, and the absorbance of the mixture was measured at 517 nm [16].

TPSC (mg/mL) was determined according to the Folin–Ciocalteu spectrophotometric method at 720 nm. First, the hardaliye samples were filtrated through a 0.45 µm syringe filter. Then, 0.2 N Folin–Ciocalteu and Na_2_CO_3_ solutions were added to the filtrated samples. The mixture was placed in a water bath adjusted to 50.0 ± 2.0 °C for 5 min. The absorbance of the samples was performed at 760 nm after the sample temperature reduced to room temperature. A calibration curve was prepared with 100, 200, 300, 400, and 500 mg/L gallic acid concentrations, and the TPSC of the samples was calculated using the equation derived from the calibration curve [16].

TMAC values (cyanidin 3-glucoside equivalent in mg/L) of the hardaliye samples were measured via the pH-differential method as a cyanidin-3-glucoside (mg/100 mL) equivalent. The samples were prepared by mixing them with 0.04 M of sodium acetate and 0.025 M of KCl, separately. The centrifugation of the mixtures was performed at 3500× *g* for 6 min, and the supernatant was taken for absorbance measurement at both 520 and 700 nm [16].

### 2.6. Inactivation of Endogenous Microflora

Total mold and yeast (TMY) and total mesophilic aerobic bacteria (TMAB) counts as log cfu/mL were performed with serial dilutions prepared with peptone water at a 0.1% concentration. The samples were subjected to surface plating with 0.1 mL of the appropriate dilutions. PCA plates for TMAB and PDA plates after acidification with 10% (*w*/*v*) tartaric acid for TMY, YPD plates for *B. bruxellensis*, and MRS plates for *L. brevis* were used. PCA, PDA, YPD, and MRS plates were incubated at 35.0 ± 2.0 °C for 24–48 h, 22.0 ± 2.0 °C for 3–5 d, 28.0 ± 2.0 °C for 5 d, and 30.0 ± 2.0 °C for 72 h, respectively [5].

### 2.7. Sensory Analyses

The hardaliye samples at room temperature were evaluated by 30 trained panelists in three phases based on a nine-point hedonic scale. They evaluated the samples for appearance (cloudiness–clarity, color intensity, dullness–shininess, and particle distribution) and then for flavor–aroma. Finally, the panelists tasted the samples in order to measure changes in important sensory attributes, such as a sour taste, bitter taste, sweetness, hardaliye taste, and aftertaste [17]. Hardaliye has a very attractive dark red color, which fades away, revealing an unpleasant brownish color with an extended storage and increased temperature. Phenolic compounds in hardaliye tend to polymerize and settle down at the bottom as the storage time and temperature increase. A sensory test for appearance was conducted to determine whether or not changes in hardaliye can be observed using visual inspection. Hardaliye has its own flavor and aroma, both of which can convert to an unpleasant aroma with a bitter and sour taste with a very strong acid aftertaste. Thus, the samples were tasted to determine whether any unpleasant taste occurred.

### 2.8. Shelf-Life Studies

The unprocessed (control) and HHP hardaliye samples (400 mL) processed under the optimal processing parameters of 490 MPa and 29 °C for 15 min were kept at 4.0 ± 0.0 and 22.0 ± 0.0 °C for 228 d for the shelf-life studies. The pH, conductivity, color (*L**, *a**, and *b**), chroma, total color difference, hue, color intensity, and inactivation of TMAB and TMY, in addition to the shininess–dullness, color intensity, clarity–cloudiness, flavor–aroma, bitter taste, sour taste, and aftertaste as sensory properties, were measured on days 0, 15, 30, 45, 66, 87, 108, 142, 180, and 228.

### 2.9. Experimental Design

The quantities and levels of the processing parameters (pressure, temperature, and treatment time) were applied based on preliminary experiments. The effects of the processing factors on titratable acidity, pH, TSS, turbidity, conductivity, reducing sugar, and color properties (*L**, *a**, *b**, chroma, total color difference, hue, color tone, color intensity, OD_420_, OD_520_, and OD_620_), as well as bioactive properties of TPSC, TAC, and TMAC and sensory properties of the cloudiness–clarity, color intensity, dullness–shininess, flavor–aroma, particle distribution, bitter taste, hardaliye taste, sweetness, sour taste, and aftertaste, in addition to microbial inactivation (TMAB, TMY, *L. brevis,* and *L. bretteromyces*) during the HHP processing of the hardaliye drink, were evaluated prior to the optimization step.

### 2.10. Optimization

The 35 responses of hardaliye, as mentioned above, were modeled as a function of pressure (X_1_, 200 to 500 MPa), temperature (X_2_, 4 to 40 °C), and treatment time (X_3_, 3 to 15 min) according to the Box–Behnken design (BBD) (Minitab version 17, Minitab Inc., State College, PA, USA) (Table 1). The quadratic regression model was thus used as the best fit to the experimental data when all the factors and interactions were significant, as follows:

Y_n_ = b_o_ + b_1_X_1_ + b_2_X_2_ + b_3_X_3_ + … + b_25_X_25_
where Y_n_ is each of the 35 response variables; X_1_, X_2_, and X_3_ are the predictors of pressure, temperature, and treatment time; and b_o_ to b_25_ are the slope coefficients, respectively. The significant terms retained in the predictive model were determined at a 95% confidence level following an analysis of variance (ANOVA). Multiple comparison tests were conducted with Tukey’s multiple comparison test. The graphical optimization was performed for the establishment of the optimum level of the three independent variables (pressure, temperature, and treatment time) to obtain the desirable responses of the maximum inactivation of *L. brevis* and minimum changes in OD_520_. Moreover, a Minitab optimizer tool was utilized to determine the optimum processing parameters of the responses.

## 3. Results

### 3.1. Changes in Properties of Hardaliye Processed with High Hydrostatic Pressure

The pH values obtained after the HHP processing varied between 3.78 ± 0.0 and 3.82 ± 0.0, while the average pH value of the control group was recorded as 3.80 ± 0.0 (Table 1). Only treatment time had a significant effect on the pH of hardaliye. The titration acidity of the control samples was determined as 5.8 ± 0.1 g/L and ranged from 4.95 ± 0.0 g/L to 5.90 ± 0.2 g/L according to the HHP processing (*p* > 0.05) (Table 1). The titratable acidity of the samples was only significantly affected by the pressure. The mean TSS values of the treated samples varied between 26.87 ± 0.1 and 27.02 ± 0.2 °Brix, whereas the mean TSS value of the control group was recorded as 27.02 ± 0.2 °Brix. Overall, no significant difference was detected between the control and treated samples for TSS (*p* > 0.05) (Table 1). The average conductivity values of the treated samples varied between 3.61 ± 0.0 and 3.78 ± 0.0 mS/cm with the average conductivity value of the control group of 3.61 ± 0.0 mS/cm. The effects of the pressure, temperature, and treatment time on the conductivity were found to be significant (*p* ≤ 0.05) (Table 1). The turbidity values of the samples ranged from 340.48 ± 1.3 to 862.89 ± 3.9 NTU, whereas the highest value was 862.89 ± 3.9 NTU in the control samples. The effects of the pressure, treatment time, and temperature alone significantly reduced the turbidity value of hardaliye (*p* ≤ 0.05) (Table 1). The mean initial reducing sugar content of the control samples was 220.32 ± 2.1 g/L. The lowest reducing sugar content was 208.32 ± 1.5 g/L, while the highest value was 253.19 ± 5.4 g/L (Table 1). The effect of the pressure and treatment time on the reducing sugar content was found to be significant (*p* ≤ 0.05), unlike the effect of the temperature (*p* > 0.05).

The mean initial *L** value was 3.33 ± 0.2, and *L** values of the treated samples varied between 2.05 ± 0.0 and 3.55 ± 0.8. The effect of pressure on the *L** value of the hardaliye drink was found to be significant (*p* ≤ 0.05), unlike the effects of the treatment time and temperature (*p* > 0.05) (Table 2). The mean initial *a** value of the control group was 8.37 ± 0.7, and *a** values of the treated samples ranged from 7.38 ± 0.3 to 10.64 ± 1.1. No significant difference was detected between the control and treated samples for *b**. The effect of the pressure on the *b** value of hardaliye was significant (*p* ≤ 0.05), whereas that of the processing time and temperature were insignificant (*p* > 0.05) (Table 2). Chroma values of the samples varied between 7.48 ± 0.3 and 10.78 ± 1.2 with the mean initial chroma value of 8.44 ± 0.7 (Table 2). The hue values of the samples varied between 0.05 ± 0.0 and 0.20 ± 0.0, whereas the hue values of the control samples were recorded as 0.12 ± 0.0 (Table 2). The effects of the processing parameters on chroma and hue values of the hardaliye drink were insignificant (*p* > 0.05). The total color difference values of the samples processed with HHP varied between 0.67 ± 0.3 and 2.08 ± 1.2 (Table 2).

The IC values of hardaliye drinks varied between 4.74 ± 0.0 and 4.87 ± 0.0, and the average IC value of the control sample was 4.85 ± 0.0. While the effects of the pressure and temperature on IC were insignificant (*p* > 0.05), the effect of the treatment time was significant (*p* ≤ 0.05) (Table 2). The color tone of the samples ranged from 0.42 ± 0.0 to 0.43 ± 0.0 with the color tone of 0.43 ± 0.0 for the control samples. No significant difference was observed between the control and treated samples for color tone, and the HHP-processing parameters exerted no significant effect on the color tone of the samples (Table 2). The control samples had the mean % OD_420_ value of 21.44 ± 0.04%, and the % OD_420_ values of the treated samples ranged from 21.36 ± 0.2 to 21.97 ± 0.2%. In general, no significant difference was observed between the control and treated samples (Table 2). The % OD_520_ values of the samples ranged from 50.21 ± 0.1 to 51.28 ± 0.2% with the % OD_520_ values of 50.41 ± 0.1% for the control samples (Table 2). Pressure, treatment time, and temperature exerted no significant effect on % OD_520_ values of hardaliye (*p* > 0.05). % OD_620_ values of the samples varied between 27.20 ± 0.1 and 28.43 ± 0.1% (Table 2). The HHP parameters did not significantly affect both O_420_ and OD_520_, whereas the OD_620_ value was significantly affected by the processing parameters (*p* ≤ 0.05).

The mean initial TPSC of the control samples, 2310.02 ± 22.9 mg/L, changed from 2222.18 ± 36.6 to 2382.24 ± 17.1 mg/L with the HHP processing. Although no significant difference was observed between the control and treated samples for TPSC, the temperature significantly impacted TPSC (Table 3). The mean initial TAC of the control samples was 70.20 ± 0.9%. The TAC of the treated samples varied between 68.91 ± 1.0 and 71.09 ± 0.9% (Table 3). The HHP-processing parameters did not significantly affect the TAC of the samples. The mean initial TMAC of the control samples was 126.91 ± 9.3 mg/L, whereas TMAC contents of the processed samples ranged from 123.25 ± 1.1 to 150.71 ± 7.4 mg/L (Table 3). The effects of the pressure and treatment time on the TMAC of the treated drink were found to be insignificant (*p* > 0.05), whereas the effect of temperature was significant (*p* ≤ 0.05).

The mean initial TMAB count of hardaliye was 5.10 ± 0.0 log cfu/mL, and the reduction in TMAB ranged from 0.46 ± 0.0 to 5.10 ± 0.0 log cfu/mL (Table 3). The pressure and temperature significantly affected the inactivation of TMAB, unlike the treatment time (*p* > 0.05). The mean initial TMY population of hardaliye was 4.21 ± 0.0 log cfu/mL. The reduction in the TMY count ranged from 0.57 ± 0.1 to 4.21 ± 0.0 log cfu/mL (Table 4). The effect of the pressure on the TMY inactivation was significant; however, the treatment time and temperature did not significantly affect the TMY inactivation. The mean initial *B. bruxellensis* count of 4.91 ± 0.6 log cfu/mL was significantly reduced by all the HHP treatments. The decrease in the *B. bruxellensis* count ranged from 0.50 ± 0.3 to 5.38 ± 0.6 (Table 4) with a significant effect produced by both pressure and temperature. The mean initial *L. brevis* population of the control samples was 5.05 ± 0.2 log cfu/mL, and the inactivation of *L. brevis* changed from 0.16 ± 0.0 to 5.05 ± 0.2 log cfu/mL after the HHP treatments (Table 4). The inactivation of *L. brevis* was significantly affected by the pressure, treatment time, and temperature.

The application of the HHP treatments caused no significant difference in the following sensory properties: the color intensity, cloudiness–clarity, particle distribution, dullness–shininess, flavor–aroma, sweetness, sour taste, bitter taste, and aftertaste. The HHP-treated samples received higher scores than the control samples did for all the measured properties (*p* > 0.05).

### 3.2. Optimization of High-Hydrostatic-Pressure Conditions for Hardaliye Drink

After the effects of the physical, chemical, microbiological, and sensorial attributes were analyzed, OD_520_ and the inactivation of *L. brevis* were optimized for the traditional drink after HHP due to the consideration of the *R*^2^, lack-of-fit value, and variance inflation factor (VIF). ANOVA results of OD_520_ and the inactivation of *L. brevis* values are given in Table 5. According to the ANOVA results, the insignificant terms were excluded from the models of the OD_520_ and inactivation of *L. brevis*. According to the polynomial regression model (Table 5), there was a positive correlation between the pressure and OD_520_ and also the treatment time and OD_520_ value. The quadratic treatment time term increased the OD_520_ value at a rate of 0.276% (*p* = 0.000), whereas the quadratic pressure and treatment time terms decreased the OD_520_ value at a rate of 0.182 (*p* = 0.002) and 0.187% (*p* = 0.001), respectively (Table 5).

The effectiveness degree of the operational conditions on the responses, such as OD_520_ and the inactivation of *L. brevis*, can be inferred from comparing the magnitudes of the coefficients of regression models. Pressure was the key driver with the highest rate increase for OD_520_ (0.148) and the inactivation of *L. brevis* (1.754) (Table 5). The *R*^2^ value indicated that 70 and 61% of the variations in the OD_520_ and inactivation of *L. brevis* values were attributed to the HHP processing conditions considered in this model, while the remaining 30 and 39% variation was the residuals (unexplained fraction), respectively. The goodness-of-fit (*R*^2^_adj_) of the models showed 0.65 and 0.58% of variations in the OD_520_ and inactivation of *L. brevis,* respectively. Lack-of-fit values for these two models were insignificant, which showed that the model fitted the experimental data well (Table 5). The operational settings were optimized to maximize the OD_520_ value and inactivation of *L. brevis*. The best solution for the multi-response optimization based on the composite desirability function is represented in Figure 1. The ideal composite desirability function was close to 1. The maximum OD_520_ (51.27) and minimum survived cells of *L. brevis* (0.0061) were obtained with the optimum operational conditions (490 MPa at 29 °C for 15 min) (Figure 2).

The effects of the HHP-processing conditions on the multiple responses (inactivation of *L. brevis* and OD_520_) were represented using the 3D surface plots. Both the magnitude of pressure and process temperature was positively correlated with the effect on the inactivation of *L. brevis* (Figure 1). The inactivation of *L. brevis* was affected by both pressure and temperature and fell with the increased pressure and temperature for 9 min. The inactivation of *L. brevis* decreased with the increased pressure under the lowest temperature (4 °C) (Figure 1). The highest OD_520_ value was obtained under 400 MPa at 22 °C. The pressure enhanced the OD_520_ value, thus justifying the significant square terms of pressure and temperature in the model (Figure 2a). The OD_520_ value increased with the temperature and treatment time (Figure 2b). The longest treatment time maximized the OD_520_ value at the highest pressure (500 MPa) (Figure 2c).

### 3.3. Shelf-Life Studies of Hardaliye Drink

The control samples kept at 4 °C spoiled after 15 d, whereas the control samples at 22 °C spoiled after 3 d. The treated samples kept at 4 and 22 °C spoiled after 228 and 108 d, respectively. Although the storage temperature did not significantly affect the HHP-treated samples, the pH of all the samples significantly fell with the increased storage time (*p* ≤ 0.05). While the pH of the control samples at 4 °C (3.69 ± 0.03) on the first day of the shelf-life studies changed to 3.52 ± 0.04, the pH of the HHP-treated samples at 4 and 22 °C (3.83 ± 0.02 and 3.77 ± 0.03) changed to 3.69 ± 0.04 and 3.69 ± 0.09 at the end of the shelf-life studies, respectively (Appendix A). The conductivity of the treated samples did not significantly differ in response to the storage temperature or time (*p* > 0.05) (Appendix A).

The color *L** value of the hardaliye samples significantly decreased with the storage time and temperature. The color *L** value of the control samples at 4 °C (11.81 ± 1.6) on the first day of shelf-life studies decreased to 2.98 ± 0.0 after 15 d, while the *L** values of the treated samples at 4 and 22 °C (11.59 ± 0.9 and 11.81 ± 1.64) decreased to 3.47 ± 0.3 and 3.13 ± 0.5 after 228 and 108 d, respectively (*p* ≤ 0.05) (Appendix A). The color *a** values of the control and treated samples at 4 °C and treated samples at 22 °C (32.96 ± 0.6, 32.75 ± 3.2, and 32.82 ± 1.9) significantly reduced to 14.04 ± 0.1, 8.47 ± 0.3, and 5.11 ± 0.5, respectively (Appendix A). The color *b** values of 12.9 ± 1.58, 12.19 ± 1.4, and 13.80 ± 2.6 for the control and treated samples at 4 °C and 22 °C on the first day of the shelf-life studies decreased to 3.26 ± 0.1, 3.03 ± 0.1, and 2.36 ± 0.2 by the end of 15, 228, and 108 d, respectively (*p* ≤ 0.05) (Appendix A).

In parallel to the *L**, *a**, and *b** values, a significant decrease in the chroma values was observed. The chroma values of 15.47 ± 2.8, 17.24 ± 1.4, and 13.89 ± 2.6 for the control and treated samples at 4 °C and treated samples at 22 °C on the first day of the shelf-life studies decreased to 12.41 ± 0.4, 9.98 ± 0.2, and 5.62 ± 0.9, respectively, after 15, 228, and 108 d (*p* ≤ 0.05) (Appendix A). The hue values of the control samples at 4 °C (0.36 ± 0.0) and treated samples at 4 (0.34 ± 0.1) and 22 °C (0.38 ± 0.0) decreased to 0.23 ± 0.01, 0.21 ± 0.0, and 0.17 ± 0.0 at the end of 15, 228, and 108 d, respectively (*p* ≤ 0.05) (Appendix A). Except for the control samples at 4 °C, no significant change was observed for the total color difference of the treated samples at 4 and 22 °C (*p* > 0.05). The color intensity of the control samples at 4 °C and treated samples at 4 and 22 °C (4.90 ± 0.4, 4.90 ± 0.4, and 4.29 ± 0.4) significantly decreased to 3.87 ± 0.0, 2.22 ± 0.1, and 3.24 ± 0.2, respectively, with the storage time. Both storage time and temperature significantly affected the *L**, *a**, and *b** values and color intensity (*p* ≤ 0.05) (Appendix A).

A significant increase occurred in both TMAB and TMY counts of all the samples with the storage time. While the treated samples at 4 and 22 °C increased from 0.00 ± 0.0 and 0.03 ± 0.0 log cfu/mL to 2.62 ± 0.1 and 3.56 ± 0.3 log cfu/mL, the TMAB of the control samples increased from 4.00 ± 0.5 log cfu/mL to 6.56 ± 0.2 log cfu/mL, respectively (*p* ≤ 0.05) (Appendix A). The TMY count of the control samples at 4 °C and treated samples at 4 and 22 °C increased from 3.33 ± 0.3, 0.00 ± 0.0, and 0.00 ± 0.0 log cfu/mL to 4.37 ± 0.2, 2.78 ± 0.1, and 2.84 ± 0.3 log cfu/mL, respectively. The increase in TMAC and TMY were significantly affected by both the increased storage time and temperature, as the samples had a higher microbial count at 22 °C than those at 4 °C (Table 6).

A significant decrease in the clarity and a significant increase in the cloudiness of the hardaliye drink were observed with the increased time and storage temperature. The clarity–cloudiness of the control samples at 4 °C and treated samples at 4 and 22 °C (9.33 ± 0.9, 9.60 ± 0.4, and 9.88 ± 0.4 reduced to 6.77 ± 0.2, 6.56 ± 0.2, and 5.94 ± 0.2, respectively (*p* ≤ 0.05) (Appendix A). Similarly, the shininess of the samples decreased with the storage temperature, revealing an increased dullness. While the shininess of the control samples at 4 °C decreased from 9.22 ± 0.4 to 5.40 ± 0.23, that of the treated samples at 4 and 22 °C decreased from 9.66 ± 0.4 and 9.33 ± 0.3 to 7.05 ± 0.3 and 7.02 ± 0.3, respectively (*p* ≤ 0.05) (Appendix A). The color intensity of all the samples significantly decreased with the storage time and temperature. The mean initial color intensity of the control samples at 4 °C and treated samples at 4 and 22 °C (7.33 ± 1.6, 8.46 ± 0.4, and 8.22 ± 0.4) reduced to 6.44 ± 0.5, 6.00 ± 0.2, and 6.14 ± 0.4, respectively (*p* ≤ 0.05) (Appendix A). The flavor–aroma of both the control and treated samples significantly decreased with the storage time. While the flavor–aroma of the control samples at 4 °C reduced from 7.03 ± 1.4 to 6.44 ± 0.5, that of the treated samples of 8.46 ± 0.4 and 8.22 ± 0.4 at 4 and 22 °C decreased to 6.00 ± 0.2 and 6.14 ± 0.4, respectively (*p* ≤ 0.05) (Appendix A). Both the bitter and sour taste of hardaliye drinks significantly increased with the storage time and temperature. The mean bitter taste of the control samples at 4 °C (4.44 ± 0.7) increased to 6.41 ± 0.6, while the treated samples at 4 and 22 °C (3.66 ± 0.7 and 4.02 ± 0.42) increased to 4.02 ± 0.3 and 4.34 ± 0.3, respectively (*p* ≤ 0.05) (Appendix A). The mean initial sour taste (4.42 ± 0.5, 3.44 ± 0.4, and 4.02 ± 0.4) for the control and treated samples at 4 °C and treated samples at 22 °C increased to 6.98 ± 0.6, 4.02 ± 0.3, and 4.34 ± 0.3, respectively (*p* ≤ 0.05) (Appendix A). The aftertaste of the control samples at 4 °C (6.78 ± 0.2) and treated samples at 22 °C (7.04 ± 0.49) were significantly reduced to 3.48 ± 0.6 and 5.67 ± 0.5, respectively, while the treated samples at 4 °C (7.84 ± 0.4) reduced to 7.26 ± 0.5 (*p* > 0.05) (Appendix A).

## 4. Discussion

The production of hardaliye may show small differences as pomace from different grape varieties can be used with different ratios of sugar and mustard seeds, whether they are whole or crushed as well as fresh or roasted. The amount of sour cherry leaves can be adjusted depending on the sensory properties of the mixture. Thus, previous studies have been conducted mostly to determine the quality properties and changes in these properties during storage, as phenolic compounds are degraded to some extent depending on the storage conditions [3]. For example, the determination of quality properties of hardaliye produced from Müşküle-type grapes with the addition of mustard seeds (1.5%) and potassium benzoate (0.1%) fermented at 30 °C revealed that the obtained drink had 17.5% TSS, 3.9 g/L of total acidity as tartaric acid, a 0.37% (*v*/*v*) alcohol content, a 0.147 g/L volatile acid content, and a pH of 4.09, with a TPSC of 272.53 mg GAE/L [18]. Properties of hardaliye may change depending on the minor differences in the production method and raw materials; thus, differences may be seen among different hardaliye samples. In particular, the grape variety plays a major role in flavor, color, and other characteristic properties of hardaliye.

Hardaliye produced from papazkarası grapes with a blue-black color stored at 4 and 20 °C for 60 d revealed the highest proportion of a red color at the beginning of the shelf-life studies; however, 60 and 78% losses in anthocyanin content were reported by the end of the shelf-life studies at 4 and 20 °C, respectively. TPSC and TAC values of the samples were measured as 1743.00 ± 8.67 mg GAE/L and 8.53 mM Trolox/mL in the fresh beverage, respectively. A higher storage temperature revealed a higher amount of an anthocyanin loss during storage with increased polymeric color values as well as other color parameters, revealing the polymerization of the anthocyanins [3]. Similar to the findings of the present study, color properties of hardaliye significantly reduced with the storage time and temperature [3].

Activation energy and activation volume are the two important factors to determine the effects of pressure and temperature on food constituents. Differences in sensitivity or resistivity toward temperature (activation energy) and pressure (activation volume) result in either the retention or the destruction of food constituents, thus altering the food structure as well as optimizing the food safety [19]. It is known that the effect of HHP on low molecular weight components and molecules is minimal, showing resistivity to activation energy and activation volume; thus, most food compounds, such as pigments, flavor compounds, and vitamins, are not adversely affected by processing parameters [20]. This fact allows for the preservation of not only nutritional value but also physicochemical properties of HHP-processed food.

Changes in some properties of HHP-processed food are related to the increased permeability and extraction yield of some compounds as well as secondary metabolites since the applied activation volume along with activation energy and exposure significantly change the cell membrane integrity. Disruption in cell membrane integrity may change the physicochemical properties, such as pH, TSS, conductivity, color, and bioactive properties, depending on the mass transfer and compounds subjected to this transfer from the cell to the surrounding environment. The application of activation energy, activation volume, and temperature in the varying magnitudes resulted in some changes in the physicochemical properties of hardaliye. Overall, the activation energy in the form of pressure positively affected the physicochemical properties. More importantly, the bioactive properties of hardaliye that were either not significantly changed or enhanced by the applied process parameters indicated the permeability of the cells during the HHP processes. In fact, an increase in the TAC of orange juice [21,22] and carrot juice [23] or no significant change in tomato juice [24] with a significant increase in TPSC as well as individual polyphenols, in addition to an increase in individual anthocyanin content, were reported in grape byproducts [25,26], longan fruit pericarp [27,28], tea leaves [19], and litchi fruit pericarp [29] after HHP treatments.

The duration and extent of the HHP treatment and the temperature exert a strong influence on microbial inactivation. Pressures above 350 MPa have a lethal effect; however, the increased treatment duration and temperature generate a synergistic effect on microbial inactivation. The microbial response to the pressure treatments also depends on the microorganism type. The HHP processing, causing morphological changes in the cell structure and cell membrane, destroys genetic materials and alters some biochemical aspects related to microbial inactivation. Inactivation on vegetative cells by HHP is considered to be the product of simultaneously occurring changes in the microbial cell [30]. In fact, the inactivation of different microorganisms, including bacteria, yeast, mold, and viruses, reveals a significant degree of inactivation.

The improvement of sensory or fresh-like properties of HHP-treated foods was also reported in fruit juices, purees, jam, and jellies owing to the increased extraction of polyphenols, antocyanins, and color pigments in addition to the inactivation of enzymes degrading the food constituents [31,32].

The shelf-life extension of traditional products is of great concern for the food industry. In fact, novel processing technologies, such as ultraviolet (UV), pulsed electric fields (PEF), and HHP, have been tested to process different products, but not hardaliye. For example, the PEF processing of traditional licorice root sherbet (LRS) based on the varying parameters of electric field strength, treatment time, and temperature revealed no significant change in most of the measured properties with a significant amount of inactivation on endogenous microflora. PEF-treated LRS samples had a shelf life of 40 d, whereas the control samples lasted for 5 d at 4 °C [16].

The HHP processing of traditional fermented turnip juice (shalgam) for 3–15 min at 4–40 °C under 200–500 MPa revealed 34.23 °C for 15 min under 500 MPa as the optimum conditions. The HHP treatment extended the shelf life over 90 d at 4 and 22 °C under the optimum parameters [5]. The HHP processing of LRS under 200–500 MPa for 3–15 min at 4–40 °C yielded the optimum operational conditions of 500 MPa for 9.90 min at 18.5 °C. Shelf-life studies conducted with the optimum HHP conditions resulted in a 25-d storage compared with the 2-to-7-d shelf life of the control samples at 4 and 22 °C [17].

In general, the HHP processing of juices resulted in no or slight changes in the physicochemical properties. For example, the HHP treatment of grapefruit juice under 600 MPa for 5 min preserved the individual antioxidants and TAC of the samples and the microbiological safety at 4 °C for 21 d [33]. The HHP processing of white grape juice concentrate in the range of 200–400 MPa for 2–4 min at 20 °C significantly reduced *Botyritis cinerea.* The TAC and total flavonoid content of HHP-treated samples significantly decreased during storage at 4 °C for 35 d [34]. The HHP processing of cloudy ginger juice presented insignificant changes in TSS, TA, pH, color, and TAC with a 3-log decrease in the microbial load. Color darkening with an increased TPSC value was reported during storage at 4 and 22 °C [35].

## 5. Conclusions

Additive-free, high-quality, and fresh-like fruit and vegetable products, in particular, fermented traditional products, such as hardaliye, attract more interest from consumers and the food industry owing to their health-promoting properties. In parallel to this trend, the novel and minimal processing of these food products has gained increasing popularity. The present study implemented HHP processing with varying levels of pressure (200–500 MPa), treatment time (3–15 min), and treatment temperature (4–40 °C) according to BBD and revealed 490 MPa at 29 °C for 15 min as the most optimal parameters for the maximized OD_520_ and inactivation levels of *L. brevis*. The shelf life of hardaliye according to these optimum parameters increased up to 228 and 108 d at 4 and 22 °C, respectively. Not only was the HHP processing effective for the preservation of the physicochemical, bioactive, and sensory properties of the hardaliye drink but it also provided a substantial inactivation of spoilage bacteria and TMY, the great hurdles to increase the shelf life of the drink. The HHP processing under 490 MPa at 29 °C for 15 min provides a possible alternative to extending the shelf life of hardaliye without the addition of any antimicrobial agents. Thus, future studies need to focus on the feasibility of HHP for hardaliye processing.

## Figures and Tables

**Figure 1 foods-12-02876-f001:**
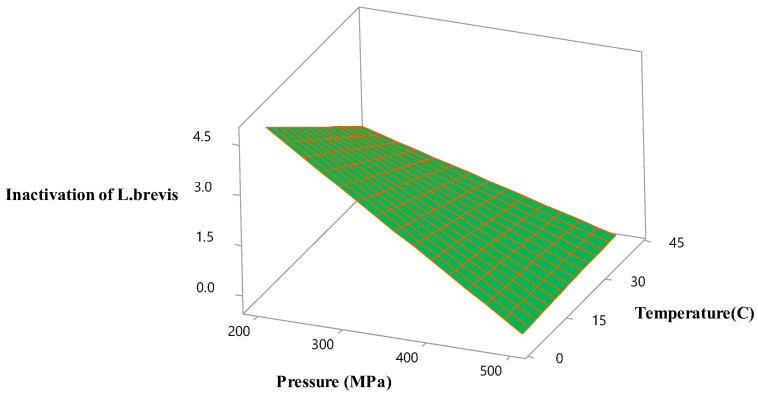
Response surface plot of the HHP parameters for the optimization of the inactivation of *Lactobacillus brevis* in hardaliye drink.

**Figure 2 foods-12-02876-f002:**
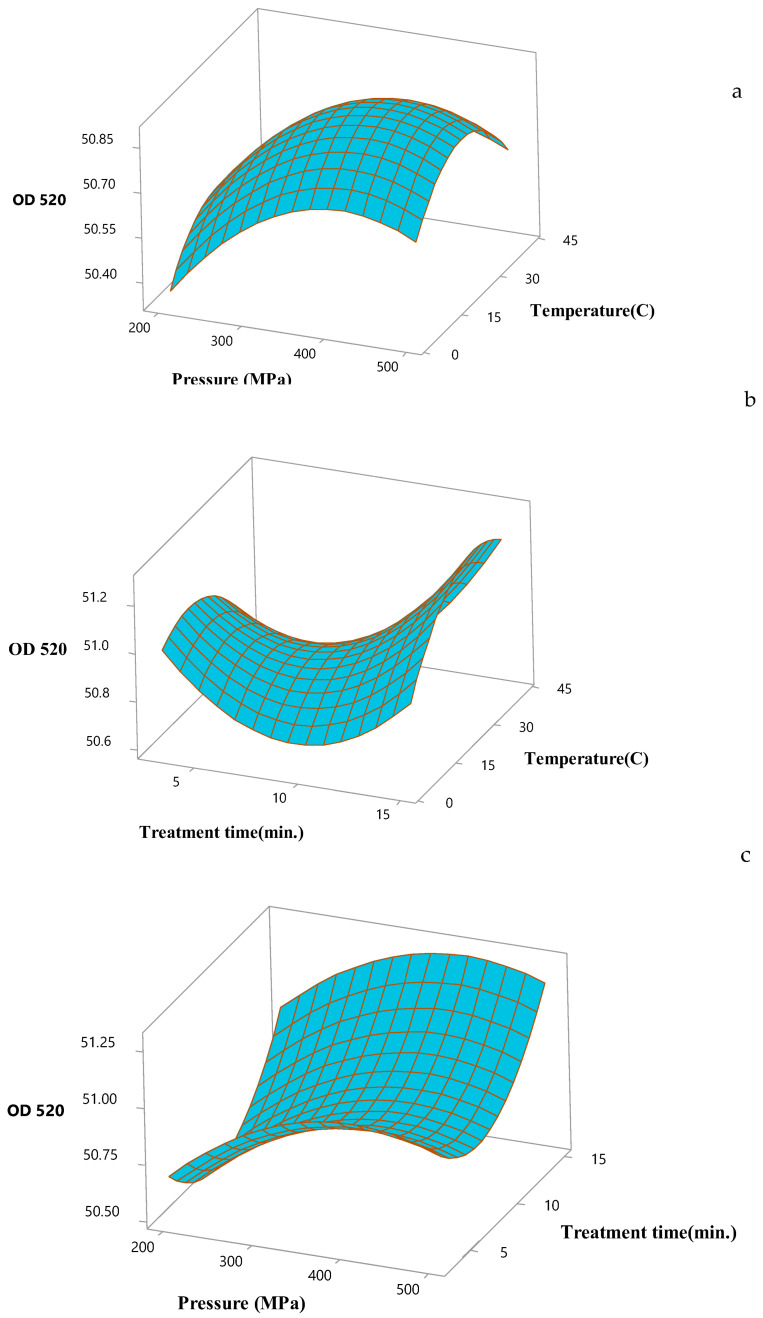
Response surface plots showing interaction effects of pressure and temperature (**a**), temperature and treatment time (**b**), pressure and treatment time (**c**) on the OD_520_ for hardaliye drink.

**Table 1 foods-12-02876-t001:** Changes in the physicochemical properties of hardaliye drink using high-hydrostatic-pressure processing with Box–Behnken design.

Process	Pressure (P, MPa)	Treatment Time (t, min)	Temperature (T, °C)	pH	Titratable Acidity (g/L)	TSS (°Brix)	Conductivity (mS/cm)	Turbidity (NTU)	Reducing Sugar (g/L)
Control	-	-	-	3.80 ± 0.0 ^ef^	5.80 ± 0.1 ^ab^	27.02 ± 0.2 ^a^	3.61 ± 0.0 ^f^	862.89 ± 3.9 ^a^	220.32 ± 2.1 ^bcde^
HHP1	350	3	40	3.78 ± 0.0 ^g^	5.35 ± 0.2 ^cdef^	27.00 ± 0.0 ^a^	3.67 ± 0.0 ^bc^	439.53 ± 2.5 ^c^	211.39 ± 7.3 ^de^
HHP2	200	3	22	3.80 ± 0.0 ^ef^	5.55 ± 0.2 ^abcde^	27.02 ± 0.1 ^a^	3.67 ± 0.0 ^bc^	359.41 ± 2.1 ^fgh^	242.50 ± 12.5 ^abc^
HHP3	350	15	40	3.81 ± 0.0 ^bcde^	5.65 ± 0.17 ^abc^	27.00 ± 0.1 ^a^	3.62 ± 0.0 ^ef^	340.48 ± 1.3 ^ı^	227.66 ± 8.6 ^abcde^
HHP4	350	9	22	3.79 ± 0.0 ^f^	5.50 ± 0.2 ^abcde^	27.02 ± 0.0 ^a^	3.69 ± 0.0 ^b^	357.27 ± 1.9 ^h^	245.41 ± 17.6 ^ab^
HHP5	200	15	22	3.80 ± 0.0 ^def^	5.65 ± 0.1 ^abc^	27.00 ± 0.0 ^a^	3.64 ± 0.0 ^cdef^	373.92 ± 2.0 ^de^	226.79 ± 5.8 ^bcde^
HHP6	350	3	4	3.80 ± 0.0 ^cdef^	5.20 ± 0.1 ^def^	27.00 ± 0.0 ^a^	3.63 ± 0.0 ^cdef^	373.84 ± 2.8 ^de^	218.99 ± 3.9 ^cde^
HHP7	500	3	22	3.80 ± 0.0 ^cdef^	5.30 ± 0.1 ^cdef^	27.00 ± 0.0 ^a^	3.65 ± 0.1 ^cdef^	369.98 ± 4.5 ^defg^	219.56 ± 3.5 ^bcde^
HHP8	350	15	4	3.81 ± 0.0 ^abcd^	5.35 ± 0.2 ^cdef^	26.87 ± 0.1 ^a^	3.62 ± 0.0 ^ef^	358.90 ± 2.9 ^gh^	208.32 ± 1.5 ^e^
HHP9	500	9	4	3.80 ± 0.0 ^cdef^	5.60 ± 0.2 ^abcd^	27.00 ± 0.0 ^a^	3.67 ± 0.0 ^bc^	482.19 ± 10.9 ^b^	253.19 ± 5.4 ^a^
HHP10	350	9	22	3.82 ± 0.0 ^ab^	4.95 ± 0.0 ^f^	27.00 ± 0.0 ^a^	3.70 ± 0.0 ^b^	363.84 ± 2.5 ^efgh^	220.43 ± 8.7 ^bcde^
HHP11	500	15	22	3.82 ± 0.0 ^a^	5.25 ± 0.2 ^cdef^	26.89 ± 0.1 ^a^	3.62 ± 0.0 ^def^	381.23 ± 1.1 ^d^	238.74 ± 11.6 ^abc^
HHP12	200	9	40	3.81 ± 0.0 ^bcdef^	5.15 ± 0.1 ^ef^	26.89 ± 0.1 ^a^	3.76 ± 0.1 ^a^	371.21 ± 3.9 ^def^	237.30 ± 5.9 ^abcd^
HHP13	200	9	4	3.81 ± 0.0 ^abc^	5.90 ± 0.2 ^a^	26.98 ± 0.0 ^a^	3.67 ± 0.0 ^bcd^	479.29 ± 5.0 ^b^	232.14 ± 3.1 ^abcde^
HHP14	500	9	40	3.81 ± 0.0 ^bcde^	5.45 ± 0.1 ^bcde^	27.00 ± 0.0 ^a^	3.78 ± 0.0 ^a^	370.57 ± 0.1 ^defg^	238.21 ± 13.4 ^abc^
HHP15	350	9	22	3.82 ± 0.0 ^a^	5.45 ± 0.2 ^bcde^	27.00 ± 0.0 ^a^	3.66 ± 0.0 ^bcde^	364.69 ± 3.8 ^efgh^	228.44 ± 8.2 ^abcde^

Data in the same column with a different superscript letter are significantly different (*p* ≤ 0.05).

**Table 2 foods-12-02876-t002:** Changes in color properties of hardaliye drink processed using high-hydrostatic-pressure processing with Box–Behnken design.

Process	*L**	*a**	*b**	Chroma	Hue	Total Color Difference	Color Intensity (IC)	Color Tone	%OD_420_	%OD_520_	%OD_620_
Control	3.33 ± 0.2 ^ab^	8.37 ± 0.8 ^b^	0.95 ± 0.3 ^abcd^	8.44 ± 0.8 ^b^	0.12 ± 0.0 ^bcde^	^―^	4.85 ± 0.0 ^ab^	0.43 ± 0.0 ^ab^	21.44 ± 0.0 ^b^	50.41 ± 0.1 ^de^	28.15 ± 0.1 ^ab^
HHP1	2.89 ± 0.3 ^abcde^	8.74 ± 0.4 ^ab^	0.63 ± 0.3 ^cd^	8.77 ± 0.5 ^ab^	0.07 ± 0.0 ^de^	0.75 ± 0.1 ^bc^	4.85 ± 0.0 ^abc^	0.43 ± 0.0 ^ab^	21.60 ± 0.1 ^ab^	50.57 ± 0.2 ^cde^	27.83 ± 0.1 ^bcd^
HHP2	2.22 ± 0.1 ^de^	8.14 ± 0.8 ^b^	1.67 ± 0.3 ^ab^	8.32 ± 0.9 ^b^	0.20 ± 0.0 ^a^	1.62 ± 0.2 ^abc^	4.79 ± 0.0 ^d^	0.43 ± 0.0 ^ab^	21.61 ± 0.1 ^ab^	50.73 ± 0.3 ^bcde^	27.65 ± 0.36 ^cdef^
HHP3	2.33 ± 0.1 ^de^	8.71 ± 1.2 ^ab^	1.71 ± 0.5 ^a^	8.89 ± 1.2 ^ab^	0.19 ± 0.0 ^ab^	1.607 ± 0.5 ^abc^	4.74 ± 0.0 ^e^	0.42 ± 0.0 ^ab^	21.48 ± 0.3 ^ab^	51.28 ± 0.2 ^a^	27.24 ± 0.2 ^fg^
HHP4	2.35 ± 0.2 ^de^	9.07 ± 1.1 ^ab^	1.73 ± 0.4 ^a^	9.24 ± 1.1 ^ab^	0.19 ± 0.0 ^ab^	1.58 ± 0.4 ^abc^	4.81 ± 0.0 ^bcd^	0.42 ± 0.0 ^ab^	21.64 ± 0.3 ^ab^	50.91 ± 0.4 ^abcd^	27.46 ± 0.2 ^defg^
HHP5	3.27 ± 0.1 ^abc^	10.64 ± 1.1 ^a^	1.74 ± 0.5 ^a^	10.78 ± 1.2 ^a^	0.16 ± 0.0 ^abc^	2.08 ± 1.2 ^a^	4.80 ± 0.0 ^cd^	0.43 ± 0.0 ^ab^	21.76 ± 0.3 ^ab^	50.89 ± 0.1 ^abcd^	27.35 ± 0.2 ^efg^
HHP6	2.29 ± 0.1 ^de^	8.08 ± 0.6 ^b^	1.47 ± 0.2 ^abc^	8.23 ± 0.6 ^b^	0.18 ± 0.0 ^ab^	1.41 ± 0.2 ^abc^	4.79 ± 0.0 ^d^	0.43 ± 0.0 ^ab^	21.78 ± 0.1 ^ab^	50.91 ± 0.1 ^abcd^	27.31 ± 0.1 ^fg^
HHP7	3.50 ± 0.5 ^a^	8.72 ± 0.8 ^ab^	0.61 ± 0.4 ^cd^	8.75 ± 0.8 ^ab^	0.07 ± 0.0 ^de^	0.93 ± 0.0 ^abc^	4.81 ± 0.0 ^cd^	0.43 ± 0.0 ^ab^	21.58 ± 0.2 ^ab^	51.09 ± 0.1 ^ab^	27.32 ± 0.1 ^fg^
HHP8	2.31 ± 0.1 ^de^	7.95 ± 0.1 ^b^	0.70 ± 0.1 ^cd^	7.99 ± 0.1 ^b^	0.09 ± 0.0 ^cde^	1.33 ± 0.1 ^abc^	4.80 ± 0.0 ^cd^	0.43 ± 0.0 ^ab^	21.64 ± 0.2 ^ab^	51.03 ± 0.2 ^abc^	27.34 ± 0.1 ^fg^
HHP9	3.55 ± 0.8 ^a^	8.36 ± 0.6 ^b^	0.99 ± 0.2 ^abcd^	8.43 ± 0.6 ^b^	0.12 ± 0.0 ^bcde^	0.95 ± 0.3 ^abc^	4.81 ± 0.0 ^bcd^	0.43 ± 0.0 ^ab^	21.53 ± 0.1 ^ab^	50.61 ± 0.2 ^bcde^	27.86 ± 0.1 ^bcd^
HHP10	2.38 ± 0.1 ^de^	8.14 ± 0.4 ^b^	0.85 ± 0.3 ^abcd^	8.19 ± 0.4 ^b^	0.10 ± 0.0 ^cde^	1.18 ± 0.3 ^abc^	4.82 ± 0.0 ^bcd^	0.42 ± 0.0 ^ab^	21.48 ± 0.0 ^ab^	50.84 ± 0.1 ^abcd^	27.68 ± 0.1 ^cdef^
HHP11	3.46 ± 0.2 ^a^	8.84 ± 0.5 ^ab^	0.43 ± 0.1 ^d^	8.85 ± 0.5 ^ab^	0.05 ± 0.0 ^e^	0.72 ± 0.2 ^bc^	4.80 ± 0.0 ^d^	0.42 ± 0.0 ^b^	21.39 ± 0.1 ^b^	51.09 ± 0.1 ^ab^	27.51 ± 0.1 ^cdefg^
HHP12	2.45 ± 0.3 ^cde^	8.82 ± 0.5 ^ab^	0.82 ± 0.2 ^bcd^	8.86 ± 0.5 ^ab^	0.09 ± 0.0 ^cde^	1.02 ± 0.2 ^abc^	4.84 ± 0.0 ^abcd^	0.42 ± 0.0 ^ab^	21.61 ± 0.2 ^ab^	50.46 ± 0.1 ^de^	27.94 ± 0.2 ^bc^
HHP13	2.97 ± 0.2 ^abcd^	8.41 ± 0.5 ^b^	1.09 ± 0.1 ^abcd^	8.48 ± 0.5 ^b^	0.13 ± 0.0 ^abcd^	0.67 ± 0.3 ^c^	4.87 ± 0.0 ^a^	0.43 ± 0.0 ^ab^	21.36 ± 0.2 ^b^	50.21 ± 0.1 ^e^	28.43 ± 0.1 ^a^
HHP14	2.05 ± 0.1 ^e^	7.38 ± 0.3 ^b^	1.18 ± 0.1 ^abcd^	7.48 ± 0.3 ^b^	0.16 ± 0.0 ^abc^	1.90 ± 0.2 ^ab^	4.84 ± 0.0 ^abcd^	0.43 ± 0.0 ^ab^	21.53 ± 0.1 ^ab^	50.68 ± 0.1 ^bcde^	27.79 ± 0.2 ^bcde^
HHP15	2.56 ± 0.1 ^bcde^	8.41 ± 0.2 ^b^	1.25 ± 0.2 ^abcd^	8.51 ± 0.2 ^b^	0.15 ± 0.0 ^abc^	0.94 ± 0.1 ^abc^	4.81 ± 0.0 ^bcd^	0.43 ± 0.0 ^a^	21.97 ± 0.2 ^a^	50.83 ± 0.2 ^abcd^	27.19.06 ^g^

Data in the same column with a different superscript letter are significantly different (*p* ≤ 0.05).

**Table 3 foods-12-02876-t003:** Changes in bioactive properties of hardaliye drink processed using high-hydrostatic-pressure processing according to a Box–Behnken design.

Process	TPSC (mg/L)	TAC (%)	TMAC (mg/L)
Control	2310.02 ± 22.9 ^abc^	70.20 ± 0.9 ^a^	126.91 ± 9.3 ^b^
HHP1	2222.18 ± 36.6 ^c^	71.09 ± 0.9 ^a^	137.21 ± 8.6 ^ab^
HHP2	2312.55 ± 25.9 ^abc^	69.80 ± 0.9 ^a^	133.03 ± 2.9 ^ab^
HHP3	2278.35 ± 14.4 ^bc^	70.03 ± 1.4 ^a^	136.42 ± 6.1 ^ab^
HHP4	2340.01 ± 32.5 ^ab^	71.06 ± 1.4 ^a^	140.04 ± 4.4 ^ab^
HHP5	2236.12 ± 12.3 ^c^	70.75 ± 0.9 ^a^	140.41 ± 2.1 ^ab^
HHP6	2332.83 ± 28.6 ^ab^	70.29 ± 0.8 ^a^	135.12 ± 2.3 ^ab^
HHP7	2348.03 ± 30.5 ^ab^	70.79 ± 0.9 ^a^	131.23 ± 9.3 ^ab^
HHP8	2351.83 ± 33.2 ^ab^	68.95 ± 0.3 ^a^	123.25 ± 1.12 ^b^
HHP9	2382.24 ± 17.1 ^a^	69.51 ± 0.8 ^a^	130.53 ± 4.3 ^ab^
HHP10	2277.93 ± 47.7 ^bc^	69.57 ± 0.9 ^a^	137.63 ± 8.9 ^ab^
HHP11	2346.76 ± 45.4 ^ab^	69.81 ± 0.7 ^a^	139.25 ± 7.9 ^ab^
HHP12	2347.61 ± 27.7 ^ab^	69.61 ± 0.9 ^a^	133.17 ± 3.6 ^ab^
HHP13	2302.84 ± 37.5 ^abc^	69.97 ± 0.9 ^a^	128.58 ± 15.2 ^ab^
HHP14	2236.12 ± 12.0 ^c^	68.91 ± 1.0 ^a^	130.67 ± 10.9 ^ab^
HHP15	2290.38 ± 18.8 ^abc^	69.07 ± 0.9 ^a^	150.71 ± 7.3 ^a^

Data in the same column with a different superscript letter are significantly different (*p* ≤ 0.05).

**Table 4 foods-12-02876-t004:** Inactivation of microbial flora in hardaliye drink processed using high-hydrostatic-pressure processing based on the Box–Behnken design.

Process	TMAB Inactivation (log cfu/mL)	TMY Inactivation (log cfu/mL)	*Brettanomyces bruxellensis* Inactivation (log cfu/mL)	*Lactobacillus brevis* Inactivation (log cfu/mL)
Control	-	-	-	-
HHP1	3.06 ± 0.1 ^d^	3.21 ± 0.0 ^b^	0.56 ± 0.4 ^e^	1.57 ± 0.2 ^cd^
HHP2	0.46 ± 0.1 ^k^	0.57 ± 0.1 ^h^	0.50 ± 0.3 ^e^	0.16 ± 0.0 ^g^
HHP3	5.10 ± 0.0 ^a^	4.21 ± 0.0 ^a^	4.36 ± 0.4 ^a^	3.94 ± 0.5 ^b^
HHP4	2.56 ± 0.1 ^ef^	2.73 ± 0.0 ^c^	0.92 ± 0.5 ^cde^	1.77 ± 0.3 ^cd^
HHP5	1.10 ± 0.0 ^j^	1.17 ± 0.1 ^g^	0.75 ± 0.4 ^de^	0.75 ± 0.2 ^efg^
HHP6	2.36 ± 0.0 ^g^	2.51 ± 0.1 ^e^	0.53 ± 0.4 ^e^	1.13 ± 0.1 ^def^
HHP7	4.10 ± 0.0 ^b^	4.21 ± 0.0 ^a^	4.38 ± 0.4 ^a^	5.05 ± 0.2 ^a^
HHP8	3.62 ± 0.0 ^c^	3.21 ± 0.0 ^b^	2.44 ± 0.2 ^b^	1.66 ± 0.6 ^cd^
HHP9	5.10 ± 0.0 ^a^	4.21 ± 0.0 ^a^	5.38 ± 0.7 ^a^	4.05 ± 0.2 ^b^
HHP10	2.56 ± 0.0 ^e^	2.73 ± 0.1 ^c^	2.04 ± 0.2 ^bc^	1.71 ± 0.7 ^cd^
HHP11	4.10 ± 0.0 ^b^	4.21 ± 0.0 ^a^	5.38 ± 0.6 ^a^	5.05 ± 0.2 ^a^
HHP12	1.76 ± 0.1 ^h^	1.77 ± 0.0 ^f^	1.07 ± 0.8 ^cde^	1.17 ± 0.2 ^de^
HHP13	1.56 ± 0.1 ^ı^	1.69 ± 0.0 ^f^	1.54 ± 0.4 ^bcde^	0.46 ± 0.2 f^g^
HHP14	5.10 ± 0.0 ^a^	4.21 ± 0.0 ^a^	1.90 ± 0.2 ^a^	5.05 ± 0.4 ^a^
HHP15	2.46 ± 0.0 ^f^	2.61 ± 0.0 ^d^	1.97 ± 0.3 ^bcd^	2.23 ± 0.3 ^c^

Data in the same column with a different superscript letter are significantly different (*p* ≤ 0.05).

**Table 5 foods-12-02876-t005:** Revised ANOVA results and estimated regression coefficients for the coded hardaliye drink according to the HHP model.

Term	OD_520_			Inactivation of *Lactobacillus brevis*		
	Coeff.	VIF	*p* Value	Coeff.	VIF	*p* Value
Regression						
Linear						
X_1_ (P)	0.148	1.00	0.000	−1.754	1.00	0.000
X_2_ (T)				−0.645	1.00	0.012
X_3_ (Trt)	0.124	1.00	0.002			
Square						
X_1_ × X_1_	−0.182	1.01	0.002			
X_2_ × X_2_	−0.187	1.01	0.001			
X_3_ × X_3_	0.276	1.01	0.000			
Interaction						
X_1_ × X_2_				0.731	1.00	0.040
X_1_ × X_3_						
X_2_ × X_3_	0.151	1.00	0.006			
Lack-of-fit		0.163			0.316	
Constant	50.86		0.000	1.608		0.000
*R* ^2^	0.70			0.61		
*R* ^2^ _(adj)_	0.65			0.58		
*R* ^2^ _(pred)_	0.58			0.54		

**Table 6 foods-12-02876-t006:** Changes in the endogenous microflora of hardaliye during shelf-life studies.

Storage Temperature					
		4 °C		22 °C	
TMAB (log cfu/mL)	Days	Control	HHP treated	Control	HHP treated
	0	4.00 ± 0.5 ^Aa^	0.00 ± 0.0 ^Bd^	4.00 ± 0.3 ^A^	0.3 ± 0.0 ^Be^
	15	6.56 ± 0.2 ^Ab^	0.47 ± 0.1 ^Cc^		1.02 ± 0.0 ^Bd^
	30		0.49 ± 0.1 ^Bc^		1.12 ± 0.2 ^Ad^
	45		0.58 ± 0.1 ^Ac^		1.31 ± 0.2 ^Ad^
	66		1.38 ± 0.2 ^Ab^		2.06 ± 0.2 ^Ac^
	87		1.40 ± 0.2 ^Ab^		2.61 ± 0.2 ^Ab^
	108		2.62 ± 0.2 ^Aa^		3.56 ± 0.3 ^Aa^
	142		2.24 ± 0.1 ^Aa^		
	180		2.31 ± 0.1 ^Aa^		
	228		2.62 ± 0.1 ^Aa^		
TMY (log cfu/mL)	Days	Control	HHP treated	Control	HHP treated
	0	3.33 ± 0.3 ^Aa^	0.00 ± 0.0 ^Be^	3.67 ± 0.4 ^A^	0.00 ± 0.0 g
	15	4.37 ± 0.2 ^Ab^	0.00 ± 0.0 ^Ce^		0.56 ± 0.0 ^Bf^
	30		0.00 ± 0.0 e		0.84 ± 0.2 ^Ae^
	45		0.38 ± 0.1 ^Ad^		1.04 ± 0.1 ^Ad^
	66		1.55 ± 0.2 ^Ac^		1.46 ± 0.1 ^Ac^
	87		1.86 ± 0.2 ^Ac^		2.02 ± 0.5 ^Ab^
	108		1.98 ± 0.2 ^Ac^		2.84 ± 0.3 ^Aa^
	142		2.12 ± 0.1 ^Ab^		
	180		2.48 ± 0.2 ^Ab^		
	228		2.78 ± 0.2 ^Aa^		

Data in the same column with a different lowercase superscript letter and data in the same row with an uppercase superscript letter are significantly different (*p* ≤ 0.05).

## Data Availability

Data will be provided to third parties upon request.

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
