# Peer review of "High-Pressure Processing of Traditional Hardaliye Drink: Effect on Quality and Shelf-Life Extension"

_foods, 2023, doi:10.3390/foods12152876_

Round 1
Reviewer 1 Report
In this paper the possibility of processing hardaliye drink by HHP with the assement of quality and shelf life extesnion is analyzed.
Theme of the work scientificaly novel.
Title of the work is adequate.
Goals are properly appointed.
Material and methods adequatly describe conducted testing.
Results and discussion section needs revision according to the remarks in pdf file.
Optimization of presented results is proprely conducted.
Conclusion section needs thorough revision.

Author Response
Manuscript ID: Foods-2477196
Manuscript title: High pressure processing of traditional hardaliye drink: Effect on quality and shelf-life extension
As the authors, we would like to thank you for the positive comments. We accepted all the comments and correct the manuscript accordingly
Reviewer 1.
Comments and Suggestions for Authors
Comment 1. In this paper the possibility of processing hardaliye drink by HHP with the assessment of quality and shelf life extension is analyzed.
Theme of the work scientifically novel.
Title of the work is adequate.
Response to comment 1. The authors would like to thank the reviewer for the positive comment.
Comment 2. Goals are properly appointed.
Response to comment 2. The goals are rewritten and more focused.
Comment 3. Material and methods adequately describe conducted testing.
Response to comment 3. Material and methods section was reorganized according to the Journal’s instructions and methods are explained in details.
Comment 4. Results and discussion section needs revision according to the remarks in pdf file.
Response to comment 4. This section is reorganized according to comments.
Comment 5. Optimization of presented results is properly conducted.
Response to comment 5. The authors would like to thank the reviewer for the positive comment.
Comment 6. Conclusion section needs thorough revision.
Response to comment 6. Conclusion section is reorganized.

Reviewer 2 Report
The authors describe the development and optiization of high pressure processing to improve shelf life of a traditional fermented beverage.
The following revisions should be considered:
- The number of tables is a bit excessive. Consider moving most of the material to supporting information showing only the most important point in the main document.
- Add line number to better facilitate peer review. The journal template is also not consistently used.
- Abstract and throughout: the decimal numbers must be aligned considering the measurement uncertainty. E.g. it makes no sense to report a value as 5.05 plus/minus 0.22. The last decimal is clearly not significant and all values should be rounded appropriately.
- Throughout: Correct the “degree” symbol (°C)
- Throughout: the English must be corrected by a native speaker
- Introduction, first paragraph, 3rd line: correct “with addition of with the addition of”
- Section 32; Check spacing in OD520 or OD 520?
see above
Author Response
As the authors, we would like to thank you for the positive comments. We accepted all the comments and correct the manuscript accordingly
Comments and Suggestions for Authors
The authors describe the development and optimization of high pressure processing to improve shelf life of a traditional fermented beverage.
The following revisions should be considered:
Comment 1- The number of tables is a bit excessive. Consider moving most of the material to supporting information showing only the most important point in the main document.
Response to comment 1-Both Table 6 and 8 were uploaded as supplementary material in order to reduce number of the tables.
Comment 2-Add line number to better facilitate peer review. The journal template is also not consistently used.
Response to comment 2- Line number was added and the manuscript is formatted.
Comment 3-Abstract and throughout: the decimal numbers must be aligned considering the measurement uncertainty. E.g. it makes no sense to report a value as 5.05 plus/minus 0.22. The last decimal is clearly not significant and all values should be rounded appropriately.
Response to comment 4- The decimal is corrected.
Comment 5-Throughout: Correct the “degree” symbol (°C)
Response to comment 6- The symbol was corrected.
Comment 7- Throughout: the English must be corrected by a native speaker
Response to comment 7-The manuscript was checked.
Comment 8-Introduction, first paragraph, 3rd line: correct “with addition of with the addition of”
Response to comment 8-It was corrected as suggested.
Comment 9-Section 32; Check spacing in OD520 or OD 520?
Response to comment 9- It is corrected as OD520

Reviewer 3 Report
First, the manuscript is not well prepared for review - it does not meet the formatting requirements according to the template provided by the Journal.
Abstract
Please check the last sentence, because according to it, HHP samples stored at 22C have a longer shelf life than samples with 4C. The authors should mention about the most important findings from physicochemical and sensorial evaluation of the samples.
Introduction
The authors state as the purpose/aim of the work a lot of elements from the scope of work. The authors should rewrite this passage giving the specific purpose of the paper and then give the scope of the research work done.
Please expand the introduction to include a second frequently tested technique based on high pressure, namely high-pressure homogenization for preservation of juices and drinks: https://doi.org/10.3390/molecules28052018 , https://doi.org/10.1007/s11947-021-02611-4
Materials and Method
2.5. Measurement of Bioactive Properties - The methodology in this section should be described in more details.
From the text it appears that 15 different variants of HPP treatment were made? Please confirm and give more emphasis to this fact, if true, in the text.
Table 1 –please delete column with proses codes - does not bring any important information since there is a similar column with similar codes next to it.
Results and Discussion
3.1. Changes in the Properties of Hardaliye Processed by High Hydrostatic Pressure - the entire chapter should be rewritten because there are only a presentation of results and a bare comparison between the different variants (highest, lowest and average values). Such a description is very difficult to read and monotonous. The authors do not explain the reasons of the changes, they do not try to explain the phenomenon of HPP parameters, and do not confront the results with the literature.
3.3. Shelf-life Studies of Hardaliye Drink- the text from page 9 and up to the middle of page 10 shows the same problem as the above mentioned section. On the other hand, the text from the middle of page 10 to page 11 should be included in the discussion of the individual determinations of the stored test beverage. Meanwhile, the authors describe in individual paragraphs again bare information only that from the literature, dumping the reader with an terribly large number of parameters. In all of this, there is no connection between the reported results of own research and those of the literature.
Table 6 and 8- I suggest moving these tables to supplementary material or extracting the most important differentiating parameters from these tables.
Table 7- please add the units in which the values are given
Conclusions
Only sentences cited below are the conclusions of the authors' research. The remaining text should be removed because it is a repetition of the given information from earlier parts of the article. The authors should expand the conclusions section to include the best HPP parameters for beverage processing and conclusions from storage studies.
“It is shown in this study that HHP provides a possible alternative to extend shelf life of hardaliye without addition of any antimicrobial agent. Thus, future studies need to focus on feasibility of HHP on hardaliye processing.”
The authors should read the entire text because there are Turkish words in many places. Many sentences throughout the text need linguistic, grammatical improvement.
Author Response
Manuscript title: High pressure processing of traditional hardaliye drink: Effect on quality and shelf-life extension
As the authors, we would like to thank you for the positive comments. We accepted all the comments and correct the manuscript accordingly.
Comments and Suggestions for Authors
First, the manuscript is not well prepared for review - it does not meet the formatting requirements according to the template provided by the Journal.
Abstract
Comment 1. Please check the last sentence, because according to it, HHP samples stored at 22C have a longer shelf life than samples with 4C. The authors should mention about the most important findings from physicochemical and sensorial evaluation of the samples.
Response to comment 1. The last sentence in abstract is corrected.
Introduction
Comment 1. The authors state as the purpose/aim of the work a lot of elements from the scope of work. The authors should rewrite this passage giving the specific purpose of the paper and then give the scope of the research work done.
Response to comment 1. The purpose of the work is reorganized and now more focused.
Comment 2. Please expand the introduction to include a second frequently tested technique based on high pressure, namely high-pressure homogenization for preservation of juices and drinks: https://doi.org/10.3390/molecules28052018 , https://doi.org/10.1007/s11947-021-02611-4
Response to comment 2. The introduction section is extended with these two recommended articles.
Materials and Method
Comment 1. 2.5. Measurement of Bioactive Properties - The methodology in this section should be described in more details.
Response to comment 1. This section is expanded and organized according to the Journal’s instructions.
Comment 2. From the text it appears that 15 different variants of HPP treatment were made? Please confirm and give more emphasis to this fact, if true, in the text.
Response to comment 2. An information indicating 15 different variants of HHP treatment was added to section 2.3 High Hydrostatic Pressure
Comment 3. Table 1 –please delete column with proses codes - does not bring any important information since there is a similar column with similar codes next to it.
Response to comment 3. The column was deleted.
Results and Discussion
Comment 1. 3.1. Changes in the Properties of Hardaliye Processed by High Hydrostatic Pressure - the entire chapter should be rewritten because there are only a presentation of results and a bare comparison between the different variants (highest, lowest and average values). Such a description is very difficult to read and monotonous. The authors do not explain the reasons of the changes, they do not try to explain the phenomenon of HPP parameters, and do not confront the results with the literature.
Response to comment 1. Effects of 15 different HHP treatments is important to understand what processing factors were effective on measured properties and in fact it was important to determine optimum processing parameters. That’s why the results section was constructed in this way. However, in parallel to this comment this section was shortened and more focused now.
Comment 2.3.3. Shelf-life Studies of Hardaliye Drink- the text from page 9 and up to the middle of page 10 shows the same problem as the above-mentioned section. On the other hand, the text from the middle of page 10 to page 11 should be included in the discussion of the individual determinations of the stored test beverage. Meanwhile, the authors describe in individual paragraphs again bare information only that from the literature, dumping the reader with an terribly large number of parameters. In all of this, there is no connection between the reported results of own research and those of the literature
Response to comment 2. This section is reorganized and more discussion in connection with literature was added.
Comment 3. Table 6 and 8- I suggest moving these tables to supplementary material or extracting the most important differentiating parameters from these tables.
Response to comment 3. Both Table 6 and 8 were uploaded as supplementary material.
Comment 4. Table 7- please add the units in which the values are given
Response to comment 4. The units were added.
Conclusions
Comment 1.Only sentences cited below are the conclusions of the authors' research. The remaining text should be removed because it is a repetition of the given information from earlier parts of the article. The authors should expand the conclusions section to include the best HPP parameters for beverage processing and conclusions from storage studies.
“It is shown in this study that HHP provides a possible alternative to extend shelf life of hardaliye without addition of any antimicrobial agent. Thus, future studies need to focus on feasibility of HHP on hardaliye processing.”
Response to comment 1. This section was reorganized.
Comments on the Quality of English Language
Comment 1. The authors should read the entire text because there are Turkish words in many places. Many sentences throughout the text need linguistic, grammatical improvement.
Response to comment 1. The whole manuscript was reread and improved grammatically with deletion of the words in Turkish.

Reviewer 4 Report
Dear Authors
the paper studied HHP to extend the shelf life of hardaliye, a tipical Tukish product, with some nutritional properties. The subject is interesting to extend the production of this beverage, but the paper needs major revisions, it should be reoganized.
It is not easy to revise the paper because there are not the number of lines.
Please, see the attached file with the observations
Abstract: it is necessary to improve the abstract following the instructions to authors. See also these observations:
1.this sentence is not clear, please improve it: Short shelf life of hardaliye limits its consumption and search in trend to process hardaliye to extend its shelf life.
2. Explain HHP the first time you cited it in the abstract.
4.OD5 Explain OD5
5. Tables and figures are not put in the text, but at the end of the paper. It is difficult to read the results.

The paper should be deeply revised for the English language.
Here are some observations:
1.Abstract: Maximum 5.10±0.00, 4.21±0.00, 5.38±0.59 and 5.05±0.22 log reductions were obtained respectively for (in) total mesophilic aerobic bacteria, total mold and yeast, Brettanomyces bruxellensis and Lactobacillus brevis by HHP.
Introduction
2.Hardaliye is a traditional non-alcoholic beverage produced by a red...using red grape pomace.....
3. The This drink is produced by lactic acid fermentation at room temperature for 7–10 days with addition of with the addition of different concentrations of whole/ground or heat-treated mustard seeds and of sour cherry leaves.
4. Its characteristics (is singular),,
5.please change this sentence:......attributed to its ingredients of grapes rich with phenolic compounds and mustard seeds containing etheric oils, allyl isothiocyanate and sinigrin which is a cinogenesis-suppressing agent.
Do you mean this? ...attributed to its ingredients: the grapes with a high phenolic content and the mustard seeds, containing etheric oils, allyl isothiocyanate and sinigrin, a cinogenesis-suppressing agent.
The others are indicated in the attached file.
Author Response
Manuscript ID: Foods-2477196
Manuscript title: High pressure processing of traditional hardaliye drink: Effect on quality and shelf-life extension
As the authors, we would like to thank you for the positive comments. We accepted all the comments and correct the manuscript accordingly
The paper studied HHP to extend the shelf life of hardaliye, a typical Turkish product, with some nutritional properties. The subject is interesting to extend the production of this beverage, but the paper needs major revisions, it should be reorganized.
Comment 1-It is not easy to revise the paper because there are not the number of lines.
Response to comment 1-Line numbering was added.
Comment 2-Please, see the attached file with the observations
Response to comment 2- The manuscript corrected accordingly.
Abstract: it is necessary to improve the abstract following the instructions to authors. See also these observations:
Comment 1- 1.this sentence is not clear, please improve it: Short shelf life of hardaliye limits its consumption and search in trend to process hardaliye to extend its shelf life.
Response to comment 1- This sentence was corrected as “…Hardaliye has very short shelf-life; thus, current efforts have been made to process hardaliye with novel processing technologies for shelf-life extension.”
Comment 2-Explain HHP the first time you cited it in the abstract.
Response to comment 2- It was explained.
Comment 3-.D5 Explain OD520
Response to comment 3- It was explained.
Comment 4- Tables and figures are not put in the text, but at the end of the paper. It is difficult to read the results.
Response to comment 4- Tables and comments were put to the end according to instructions.
Comments on the Quality of English Language
Comment 1-The paper should be deeply revised for the English language.
Here are some observations:
1.Abstract: Maximum 5.10±0.00, 4.21±0.00, 5.38±0.59 and 5.05±0.22 log reductions were obtained respectively for (in) total mesophilic aerobic bacteria, total mold and yeast, Brettanomyces bruxellensis and Lactobacillus brevis by HHP.
Introduction
2.Hardaliye is a traditional non-alcoholic beverage produced by a red...using red grape pomace.....
- TheThis drink is produced by lactic acid fermentation at room temperature for 7–10 days with addition of with the addition of different concentrations of whole/ground or heat-treated mustard seeds and of sour cherry leaves.
- Its characteristics (is singular),,
5.please change this sentence:......attributed to its ingredients of grapes rich with phenolic compounds and mustard seeds containing etheric oils, allyl isothiocyanate and sinigrin which is a cinogenesis-suppressing agent.
Do you mean this? ...attributed to its ingredients: the grapes with a high phenolic content and the mustard seeds, containing etheric oils, allyl isothiocyanate and sinigrin, a cinogenesis-suppressing agent.
The others are indicated in the attached file.
Response to comment1-All the suggested corrections were made.

Round 2
Reviewer 1 Report
Paper is sufficiently improved to be accepted
Author Response
Manuscript ID: Foods-2477196
Manuscript title: High pressure processing of traditional hardaliye drink: Effect on quality and shelf-life extension
As the authors, we would like to thank you for the positive comments. We are glad that the manuscript is found satisfactory to be published.

Reviewer 3 Report
The authors have made a number of very important improvements. The most important is to rewrite the abstract and conclusions in a clearer and more concrete way. The conclusions are now factual, detailed and based on the authors' research findings. The authors broke down the presentation of the results into a results and a discussion section, which allowed better clarity of the message.
Most importantly, they significantly improved the formatting of the text and the English language.
I am glad that the authors decided to transfer some of the results to the supplementary material for better presentation of results.
However, I still have a few comments on the implementation of the corrections previously indicated, as well as some new ones:
1) I asked the authors to briefly expand in the introduction that among the techniques based on high pressure is also high-pressure homogenization. The authors wrote in response to comments that they mentioned this in the introduction but they did not. I still think this is an important issue in the context of liquid foods preserved by high-pressure techniques.
2) Combine the paragraph at the beginning of the discussion (lines 524-538) with the text from the introduction. It deals with introducing the characteristics of the product to the reader, so it should be in the introduction section.
3) 2.4. Measurement of Physicochemical Properties - Please describe the conditions for color analysis in more detail (light source, observer angle, mode of reflection or transition). Please also expand the abbreviation OD.
Author Response
Manuscript ID: Foods-2477196
Manuscript title: High pressure processing of traditional hardaliye drink: Effect on quality and shelf-life extension
As the authors, we would like to thank you for the positive comments. We accepted all the comments and correct the manuscript accordingly
Reviewer 3.
The authors have made a number of very important improvements. The most important is to rewrite the abstract and conclusions in a clearer and more concrete way. The conclusions are now factual, detailed and based on the authors' research findings. The authors broke down the presentation of the results into a result and a discussion section, which allowed better clarity of the message.
Most importantly, they significantly improved the formatting of the text and the English language.
I am glad that the authors decided to transfer some of the results to the supplementary material for better presentation of results.
However, I still have a few comments on the implementation of the corrections previously indicated, as well as some new ones:
Comment 1) I asked the authors to briefly expand in the introduction that among the techniques based on high pressure is also high-pressure homogenization. The authors wrote in response to comments that they mentioned this in the introduction but they did not. I still think this is an important issue in the context of liquid foods preserved by high-pressure techniques.
Response to comment 1) High pressure processing and high-pressure homogenization are to complete processes. Thus, we explained HHP in more detail. Now, we extended introduction section more including high pressure homogenization with the following paragraph:
“Application of high pressure in the range of 20-100 MPa with homogenization known as high pressure homogenization (HPH) is also common application to process foods products, especially juices and dairy beverages with the main objectives of particle size reduction and increase emulsion stability by preventing coalescence phenomena and creaming. Moreover, the magnitude of pressure has direct effect on cell disruption and recovery of intracellular bioactive compounds which provides this technology to be used for food processing with improvement of food safety and shelf-life [8,9].”
Comment 2) Combine the paragraph at the beginning of the discussion (lines 524-538) with the text from the introduction. It deals with introducing the characteristics of the product to the reader, so it should be in the introduction section.
Response to comment 2) The paragraph at the beginning of the discussion was moved to introduction section as suggested.
Comment 3) 2.4. Measurement of Physicochemical Properties - Please describe the conditions for color analysis in more detail (light source, observer angle, mode of reflection or transition). Please also expand the abbreviation OD.
Response to comment 3) The following information was added to the color measurement section.
“…D65/10° as simulation of daylight color is standard illuminant with 10 degrees viewing angle was used for color measurement. A light source used for color measurement emits radiant energy in the form of visible light, a minor portion of the electromagnetic spectrum including ultraviolet, X-rays, radio waves, and infrared light in the range of 400-700 nm. Absorbance mode was utilized for the color measurements.”
Regarding the OD, the sentence was arranged as …..calculated from the optical density (OD) measurements at 620, 520, and 420 nm, and reported as RCT (OD620), BCT (OD520), and YCT (OD420), respectively.”

Reviewer 4 Report
Dear Authors,
the paper needs only some minor revisions.
Line 15: This drink is production: remove is
Line 31: consequently..do you mean respectively?
Line 42: If fermentation practiced at...
change in . If fermentation is practiced at
Line 50-54: please improve this sentence, here is an example:
Grapes and fermentation process also help to develop nutritional value of hardaliye, whereas the functional and health promoting properties are attributed to its ingredients. In particular, the grapes which have a high phenolic content and the mustard seeds, containing etheric oils, allyl isothiocyanate and sinigrin, a cinogenesis-suppressing agent.
Line 173: .... parameters were; change in .... parameters were:
Author Response
Manuscript ID: Foods-2477196
Manuscript title: High pressure processing of traditional hardaliye drink: Effect on quality and shelf-life extension
As the authors, we would like to thank you for the positive comments. We accepted all the comments and correct the manuscript accordingly
Reviewer 4.
Dear Authors,
the paper needs only some minor revisions.
Comment 1-Line 15: This drink is production: remove is
Response to comment 1- It is removed.
Comment 2-Line 31: consequently..do you mean respectively?
Response to comment 2- It is corrected as respectively.
Comment 3-Line 42: If fermentation practiced at... change in If fermentation is practiced at
Response to comment 3- It is corrected as suggested.
Comment 4-Line 50-54: please improve this sentence, here is an example: Grapes and fermentation process also help to develop nutritional value of hardaliye, whereas the functional and health promoting properties are attributed to its ingredients. In particular, the grapes which have a high phenolic content and the mustard seeds, containing etheric oils, allyl isothiocyanate and sinigrin, a cinogenesis-suppressing agent
Response to comment 4- The sentence is corrected as suggested.
Comment 5-Line 173: .... parameters were; change in .... parameters were:
Response to comment 5-It is corrected as suggested.
